# Forest Quality and Available Hostplant Abundance Limit the Canopy Butterfly of *Teinopalpus aureus*

**DOI:** 10.3390/insects13121082

**Published:** 2022-11-24

**Authors:** Lu Wang, Hui Wang, Yuhang Zha, Heyi Wei, Fusheng Chen, Juping Zeng

**Affiliations:** 1Key Laboratory of National Forestry and Grass and Administration on Forest Ecosystem Protection and Restoration of Poyang Lake Watershed, College of Forestry, Jiangxi Agricultural University, Nanchang 330045, China; 2The Station of Observation and Research of Jiulianshan, Longnan 341701, China; 3Jiulianshan National Nature Reserve of Jiangxi, Longnan 341701, China; 4Geodesign Research Centre, Jiangxi Normal University, Nanchang 330022, China

**Keywords:** canopy butterfly, Magnoliaceae, hostplant limitation, habitat selection, hostplant availability

## Abstract

**Simple Summary:**

We carried out multi-year investigations on habitat selection of early stage individuals of the forest canopy species *Teinopalpus aureus* in Jiulianshan, South China, by comparing the low-mountain and middle-mountain regions as well as three types of forest: primeval broadleaf forest, secondary broadleaf forest, and coniferous forest. It was found that this butterfly exclusively occurred in the middle-mountain region and only preferred primeval broadleaf forests. This could mainly be driven by the specific larval hostplants (i.e., three Magnoliaceae species). Such resources were superior in the middle-mountain region, including plant abundance, diversity, tree height, DBH (Diameter at breast height), etc., and such a resource advantage was more concentrated in the primeval broadleaf forests. In particular, the abundance and diversity of hostplant trees with an exposed crown, which is demanded by this butterfly in its oviposition and in the subsequent larval development, were higher in the primeval broadleaf forests. Therefore, both the forest quality and the availability of the hostplants together limited the occurrence of this canopy butterfly.

**Abstract:**

Hostplant limitation is a key focus of the spatial interaction between a phytophagous butterfly and a hostplant. The possible drivers related to the hostplants are species richness, abundance, or availability, but no consensus has been reached. In this study, we investigated the butterfly–hostplant interaction using the case of the forest canopy butterfly *T. aureus* in Asia, whose narrow distribution is assumed to be limited by its exclusive hostplant, Magnoliaceae, in tropic and subtropic regions. We recorded the Magnoliaceae species, as well as plant and butterfly individuals in transect, and we collected tree traits and topography variables. The results confirm that this butterfly is limited by the hostplants of their larval stage. The hostplants occurred exclusively in the middle-mountain region, with preference only for primeval forests. The hostplant resource was superior in the middle-mountain region, particularly concentrating in primeval forests. The hostplant’s abundance, together with altitude and habitat types, was critical to this butterfly’s occurrence, while those hostplant trees with an exposed crown, which are demanded by this butterfly in its oviposition, were the best drivers of positive butterfly–hostplant interactions. Therefore, the hostplant’s limitation was mainly determined by the availability of the hostplant. This case study supports the hypothesis that the limitation on this butterfly’s occurrence was driven by the hostplant’s availability, and it suggests that protecting high-quality forests is a valuable activity and essential in the conservation of canopy butterflies.

## 1. Introduction

Hostplant limitation (HPL) is a key focus of the spatial interaction between a phytophagous butterfly and a hostplant [1,2,3,4], and it can be affected by the (1) presence or absence of the hostplant [5], (2) the hostplant’s abundance (HPAb) and species richness (HPSR) [6], and (3) the hostplant’s availability (HPAv) [7,8,9]; however, a consensus is still lacking thus far in this regard. In this study, by using the oligophagous butterfly case of *Teinopalpus aureus* in Jiulianshan, South China, we attempted to illustrate the roles of hostplant abundance (HPAb), hostplant species diversity (HPSD), and hostplant availability (HPAv) in the formation of the hostplant limitation.

The accelerated decline in biodiversity with global warming [10,11] and the changes in the distribution and dynamics of the population of butterflies (as sensitive indicators) have attracted significant attention throughout the world. For oligophagous and monophagous butterflies, it is critical to know their distributions and the changes in their HPLs. They have a relatively narrow diet width, being dependent on their hostplants, and in terms of distribution, they are more likely to be limited by both abiotic (climate, topography, etc.) [3] and biotic factors (e.g., bottom-up control by hostplants) [12]. Thus, in nature, these specialists often occur in narrow, specific regions [5] corresponding to the constraint from the fundamental niche to the realized one [13]. Furthermore, future climate changes will certainly affect the ability of butterfly populations to persist, especially in cases where plants and butterflies react asynchronously [14,15], which could cause a spatial mismatch between butterfly and hostplant distributions [1,3]. This will undoubtedly lead to further deterioration of the situation of these specialists [16], leading to accelerated species extinction and a rapid decline in regional diversity [17]. As such, strategically establishing their key habitat as a protection area is a basic element in the conservation of these populations [18]. According to the functional concept of “resource-based habitats” [19,20,21], it is necessary to focus on the key consuming resources (e.g., hostplants) associated with habitat conservation and to conduct field investigations to determine the basic habitat demands and occurrence patterns under the effect of an HPL in order to provide efficient information (e.g., prioritizing resources or locations) for conservation-related decisions [22].

Mountains are key to habitat conservation in this changing world, given their coverage of a wide spectrum of environmental conditions, particularly vertical gradients [23]. In Asia’s tropical and subtropical regions, mountains are widely distributed and forests are usually lush, developing continuously from low to high elevations along vertical gradients with distinct layer structures. These mountain forests provide a variety of plant resources and microhabitats for phytophagous butterflies, conserving most of the richness of large swallowtail species [24]. Most species are endemic to this region, including the two endemic butterflies *Byasa impediens* [25] and *Bhutanitis thaidina* [26], and they always explore their favorite microhabitats and form specific preferences for long-term adaptation. For example, butterflies from the species *Teinopalpus*, *Meandrusa*, *Pazala*, *Agehana*, *Graphium*, etc. prefer to stay in the unique evergreen broadleaved canopies of mountain forests throughout their early stages, where the larvae usually feed on the treetop leaves of certain evergreen trees (Lauraceae, Magnoliaceae species, etc. [24,27,28]) because these trees provide favorable sunshine conditions for the incubation of their large-sized eggs. Therefore, it is assumed that these canopy butterflies can be constrained by their hostplants (e.g., HPL) and the forest canopy quality, and their actual distributions may be further narrowed by the combined effect of these factors. However, considering the uncertain active behaviors and range during the adult stages of these species [19,20,21], this study only focused on the relationship between the butterflies’ early stages (i.e., egg and larva) and the hostplants. We expected to analyze the HPL based on the clear resource-based habitat quality as well as the characteristics involved.

*Teinopalpus aureus* is one of the largest swallowtail butterflies that is endemic to tropical and subtropical regions in Asia [24,27,28]. It seems to be constrained to mountains with steep terrain and primeval forest cover [27,29,30,31,32]. It is rare to encounter this species in the field, but it is known as a flagship species [33] and has received close attention from entomologists, environmentalists, and the public [34,35,36]. However, it is still listed as a DD (data-deficient) species by the IUCN (International Union for Conservation of Nature), and it is still unclear what indeed constrains its occurrence. The two sister species of the genus *Teinopalpus*, *T. aureus* and *T. imperialis* Hope, occur only in some narrow mountain areas. They both prefer to live in treetops during their early stages of life (including the egg and larval stages) and seem to depend on evergreen broad-leaved forests [37,38], since the known hostplants are both from Magnoliaceae, a dominant family in these mountain forests. Obviously, due to being canopy butterflies, fieldwork on these species is challenging. For example, Igarashi [38] first investigated *T. imperialis* in Nepal and northern India in 1962, but it was not until 1986 that the author made a breakthrough in northern India and successfully identified the first larval hostplant of *Magnolia campbellii* Hook, revealing the biological characteristics of *T. imperialis* in the field [38]. The mysterious veil of *T. aureus* was not uncovered until 2004 by Zeng [39] in the Dayaoshan Mountains in Guangxi Province, South China, where the author successfully identified two hostplants, *Parakmeria nitida* (W.W. Smith) Law and *Michelia chapensis* Dandy, through the challenging method of “climbing up trees artificially and then checking leaves one by one in the canopy”. The butterfly’s biological characteristics were also revealed [30,40]. Subsequently, by using Zeng’s challenging method, some hostplants of Magnoliaceae in *T. aureus* were also identified through several other field investigations, including *M. foveolata* Merr. ex Dandy and *M. maudiae* Dunn in the Jiulianshan Mountains [31], *M. foveolata* in the Jinggangshan Mountains [40], as well as *M. maudiae* in the Pingshan Mountains [41]. Therefore, the two oligophagous butterflies, *T. aureus* and *T. imperialis*, are assumed to be consistent with the Magnoliaceae species in terms of the local distribution and occurrence, for example, involving the HPL effect.

To collect sufficient field data for the analysis of the HPL effect on *T. aureus*, we used the ready-made case of Jiulianshan, South China, where monitoring data have been collected during the butterfly flight period since 2004. Based on previous records, the general range of adult activity in Jiulianshan was determined, to be the same as for other geographical populations [30,40,41], this butterfly was also shown to go through two generations per year in this area [31]. Moreover, as for this oligophagous butterfly, previous local efforts have identified three larval hostplants: *M. foveolata*, *M. maudiae* [31], and *Manglietia fordiana* Oliv, all belonging to Magnoliaceae. So, in the case of Jiulianshan, the existing data or works suggest that further studies related to the spatial interaction between this rare oligophagous butterfly and the known hostplants or the possible HPL effect should be conducted. This time, field investigations were carried out along previous transect lines as well as along some new lines to record Magnoliaceae species, the individual locations (longitude, latitude, altitude, etc.) of different butterfly stages (egg, larvae, and adult), and the abovedescribed hostplants. We aimed to illustrate the butterfly’s habitat requirements and the HPL effect based on three resource levels: Magnoliaceae plants, three hostplant species, and the available hostplants. Through applying the resource-based habitat concept, we attempted to determine: (1) the factors impacting the habitat selection of *T. aureus*, (2) whether the HPL effect exists, and (3) how it impacts and constrains the butterfly’s local occurrence, for example, based on resource abundance (e.g., HPAb), diversity (e.g., HPSD), or availability (e.g., HPAv).

## 2. Materials and Methods

### 2.1. Study Area

This study was conducted in the Jiulianshan Mountains, which are the eastern core part (E 114°22′50″–114°31′32″, N 24°29′18″–24°38′55″) of the Nanling Mountain Range, located in the Jiulianshan National Nature Reserve on the boundary between Jiangxi and Guang-dong Province in South China. The elevation in this study area ranges from 280 to 1430 m in altitude and is mainly composed of medium and low mountains (Figure 1). Under the combined influence of continental and marine climates as well as the effect of mountainous terrain, this area is warm and humid. Mountain forests develop well and stay green throughout the year under good hydrothermal conditions.

As a National Nature Reserve, this area mainly focuses on protecting the typical subtropical evergreen broad-leaved forests and biodiversity [42]. However, due to various forms of human interference throughout history, several types of forest are currently present, including (1) primeval forests, for example, evergreen deciduous broad-leaved mixed forests and montane dwarf forests, which have been subjected to early protection actions (since 1957) and low interference and have retained their ecological integrity [42]; (2) secondary forests, which are recovering areas that have undergone selective cutting or small-area deforestation and still have incomplete ecological integrity, for example, secondary evergreen broad-leaved forests; (3) artificial forests, mainly pine (*Pinus massoniana*) or Chinese fir (*Cunninghamia lanceolata*) coniferous forests that were planted artificially in areas undergoing large-area deforestation; (4) bamboo forests, which are areas dominated by Moso bamboo (*Phyllostachys eduli*); and (5) other open forests, such as the forest areas around farmlands and villages [42]. In previous field observations, it was found that *T. aureus* almost always appears in close forests [30,31,37,40,41], so we did not use the transect data from bamboo and open forests in the following analysis. That is, we mainly focused on the differences among the primeval forests, secondary forests, and coniferous forests.

### 2.2. Investigations of T. aureus and Magnoliaceae Trees

From 2018 to 2021, we carried out transect counting randomly along mountain roads, forest trails, ridges, and streams in the study area (Figure 1), and the GPS data for all Magnoliaceae species, tree, and butterfly individuals within 20 m on both sides of transects were recorded, including the tree DBH and height. As for the butterfly, we recorded current as well as the previous data sourced from Lin et al. [31]. In our records, we marked the individual trees of known hostplants if they contained butterfly eggs or larvae. Field investigations were conducted throughout the year, while butterfly data were collected from April to June during the first generation, as well as from August to October during the second generation [30,31,37].

In the butterfly investigations, we used direct and indirect investigation methods to record eggs or larvae in the tree canopy of Magnoliaceae. In the direct method, we checked every leaf reached after climbing up trees or by lowering tree branches using a hook. In the indirect method, we checked all suspected leaf-feeding marks and feces of larvae on the ground under the canopy of known hostplant trees, according to the descriptions presented in Zeng [39] and Lin et al. [31].

### 2.3. Calculation of the α Diversity Index and Determination of the Transect Length Unit

We recorded six Magnoliaceae species in the study area: *Michelia skinneriana* Dunn, *M. chapensis*, *M. fordiana* (including *Manglietia yuyuanensis* Law), *Magnolia odora* (Chun) Figlar & Noot., *M. maudiae,* and *M. foveolata* [43]. We used the formula H′ = −∑P_i_ LnP_i_ (where P_i_ is the relative abundance of species) to calculate the Shannon–Wiener index as the α diversity index of the transect line. We first calculated cumulative values, 10 m in length, in three typical transect lines that extended from the foot of the mountain to the top. Then, we fitted three logistic response curves for these values against the transect length (Figure 2). The results indicated that, as the length increased, the growth of the index values first accelerated and then reached points of deceleration inflection (e.g., 188 m, 226 m, and 295 m, respectively) and entered a stable status. As such, in order to ensure the stability of the index, we used 300 m as the transect length unit in the following analysis in accordance with the method of Ye et al. [44]. Lastly, we divided all the transect lines into a certain number of 300 m sections and ensured each section contained only one type of habitat and was <100 m in relative elevation, as well as having an interval of >50 m from other sections. We did not use transect sections of less than 300 m in the analysis.

### 2.4. Environmental Data Collection

Considering that the butterfly *T. aureus* requires a mountain topography for its occurrence [45], we collected DEM data with a resolution of 30 m from www.usgs.gov (accessed on 30 March 2020) and used these to generate elevation, slope, and aspect raster data in ArcGIS 10.5 (Esri, Redlands, CA, USA). We further calculated the slope direction index of TRASP using the formula TRASP = {1 − cos[(π/180) × (Aspect−30)]}/2. This index has values from 0 to 1, indicating a change in environment from wet and cold to dry and hot conditions [46]. We also collected data on two variables, tree age and forest type, from the local Forest Resource Inventory (FRI) database.

Previous field investigations showed that *T. aureus* males always prefer to be active on certain mountaintops, where they are usually waiting for females or mating chances [37]. These mountaintops are almost always above 1000 m in altitude. These behaviors and situation were also observed in Jiulianshan [31]. Therefore, we sorted transect sections into two groups, low-mountain transects (LMT) and the middle-mountain transects (MMT) (Figure 1), according to the regional elevation of mountaintops. Furthermore, we sorted the MMT areas into the occupied areas (OA), where *T. aureus* (adults and early stages) was recorded each year, and the non-occupied areas (NOA), where the butterfly was never observed (Figure 1) [19].

### 2.5. Hostplant Availability Definition

It was found that female *T. aureus* butterflies almost always lay eggs on sites with tree canopies exposed, as shown in previous observations, maybe due to the more favorable sunshine conditions [41]. Therefore, for the three known hostplants in Jiulianshan, *M. fordiana*, *M. foveolata*, and *M. maudiae*, we roughly classified the tree individuals into available and non-available groups by comparing their heights with the average tree height and defined availability as hostplant trees taller than the average height. Subsequently, we calculated the abundance and diversity indices of the available plants, in the same way as for the other two plant resource levels: Magnoliaceae plants and hostplants.

### 2.6. Statistical Analysis

We used the non-metric multidimensional scaling analysis (NMDS) method to evaluate the contributions of the variables (see Table 1) to the distributions of LMT and MMT sections in descending dimensions [45]. We further used the Mann–Whitney U test method to compare the differences among variables between LMT and MMT as well as between occupied areas and non-occupied areas. We used Bailey’s method to determine factors affecting butterfly habitat selection (such as preference, avoidance, or random) in three types of forest, primeval forest, secondary forest, and coniferous forest, by comparing the actual (P_i_) and expected (P_io_) proportions utilized. The differences in environmental variables were also compared among these three habitat types using the Kruskal–Wallis method. Lastly, for the variables with significant differences between occupied areas and non-occupied areas, we used the stepwise discriminant analysis to identify key or driving variables or factors associated with butterfly habitat selection. In this study, we did not use adult records, and the level of significance was set as *p* < 0.05. All statistics were conducted in SPSS 25 (IBM, Armonk, NY, USA) and R 4.2.1 [47].

## 3. Results

### 3.1. Distribution of T. aureus Driven by the Resource Quality of Hostplants

Figure 3 shows an interpretive ranking (Stress = 0.076, *R*^2^ = 0.98) in the descending dimension space by NMDS based on the 13 environmental variables presented in Table 1. A comprehensive gradient change in hostplant quality is presented comprehensively using MDS1 and MDS2, with the biggest contributors being identified as the AHPSD (Available hostplants’ species diversity), HPSD (all hostplants’ species diversity), MSD (all Magnoliaceae species diversity), AHPAb (Available hostplants’ abundance), and HPAb (all hostplants’ abundance) (Figure 3A). The transect sections of the early butterfly stages (e.g., the occupied MMT in Figure 3B) were significantly more inclined to be located at the high-quality side of a gradient in terms of distribution compared to others.

### 3.2. Habitat Preference and Requirements of T. aureus in Terms of Occurrence

Using Bailey’s method, we compared the actual and expected proportions utilized among three habitat types: primeval forests, secondary forests, and coniferous forests. The results in Table 2 show that in the early stages, the butterflies prefer primeval forests and avoid secondary forests and coniferous forests for habitat selection (Table 2). The comparison of 12 environmental variables between occurrence and non-occurrence transects of *T. aureus* larvae shows significant differences for 10 variables but not for the 2 topography variables (aspect and slope) (Table 3). Overall, occupied transects have higher altitudes, older tree ages, and larger Magnoliaceae plants than non-occupied transects, and they have richer and more diverse Magnoliaceae resource conditions (including hostplant resources and available hostplant resources) (Table 3).

### 3.3. Distribution and Drivers in T. aureus

Using the significant variables presented in Table 2 and Table 3 to conduct a stepwise discriminant analysis, we constructed an optimal model with four selected environmental variables: AHPAb, Altitude, Habitat, and HPAb. The discriminant accuracy was 95% in this model. A positive relationship was observed between the former three variables and butterfly occurrence, while a negative one was found for HPAb, according to the discriminant coefficients and the Wilk’s λ results presented in Table 4. AHPAb was identified as the most critical variable or the key driver in the distribution of the early stages of *T. aureus*.

As for the effect of altitude on butterflies, it was shown that better resource conditions for Magnoliaceae plants exist in MMT, including plant abundance and diversity, as well as functional traits such as DBH, tree height, and tree age (Figure 4). The better resource conditions were also shown to exist in OA but did not include the condition of plant abundance, which was not found to be significantly different between OA and NOA (Figure 5).

Similarly, as for the influences of habitat, we found that the best resource conditions for Magnoliaceae only existed in primeval forests. These were therefore the preferred habitat over secondary forests and coniferous forests (Figure 6). To be noted, although no significant difference existed regarding the abundance index for all Magnoliaceae plants and all hostplants between primeval forests and secondary forests, with the same situation occurring for the unavailable hostplants (*p* = 0.93), a significant difference was found for the available portion of hostplants (Figure 6). In primeval forests, the abundance was strongly associated with the diversity in this part, while such a relationship did not occur in all Magnoliaceae plants or in all hostplants (Figure 7).

## 4. Discussion

Phytophagous butterflies, especially oligophagous and monophagous species, are constrained in terms of their occurrence and distribution by “hostplant limitations (HPLs)” [48]. However, whether HPL is related to hostplant abundance (HPAb) and hostplant species richness (HPSR) [6] or is driven by hostplant availability (HPAv) [8,9] remains uncertain. This case study, conducted in the Jiulianshan Mountains in South China, confirmed the biological interaction between the oligophagous butterfly *T. aureus* and its exclusive Magnoliaceae hostplants. The results support the idea that the HPL is mainly driven by hostplant availability (in abundance and diversity) rather than simply by hostplant abundance. Actually, we identified a negative interaction between butterfly and hostplant abundance and a strongly positive one between butterfly and hostplant availability.

We used the NMDS method to construct a descending dimension space, which expressed a comprehensive gradient change in the hostplants’ quality, and clearly showed that *T. aureus* butterflies are inclined to occur in the high-quality side of the area (see Figure 3). The quality of an area was determined by the hostplant abundance (e.g., HPAb, MPAb, and AHPAb) and diversity (e.g., HPSR, including MSD, HPSD, and AHPSD). The comparison of occurrence and non-occurrence transects also showed that most indices of occurrence transects were better than those of non-occurrence transects. These results illustrate the strict requirements of host plant resources for the canopy butterfly *T. aureus*.

However, according to the functional concept of “resource-based habitats” [19,20,21], hostplants in a region cannot be used completely by butterflies, because if the hostplants are considered to be key consumables, at least two parts can be included: (1) the available part, which is findable (to be encountered) and utilizable in the wild and has a real function (i.e., availability), and (2) the unavailable part, which does not show any function, either temporally or spatially. This distinction could be necessary for the understanding of HPL. Most butterflies usually demand high-quality habitats during their early life stages (e.g., egg, larva), including suitable microhabitat conditions [25,26,49,50] and utilities related to their hostplants. These requirements may be associated with some specific functions, such as an increase in the release of a key stimulus, namely, oviposition-induced volatiles, with increased light intensity [51], which directly determine the availability of hostplants for butterflies, especially the specialists. Similarly, in the case of *T. aureus*, the high demand for the oviposition position is reflected by the low-density and scattered pattern of egg distribution in the canopy [41], and our results show that the availability of Magnoliaceae hostplants can be determined by the requirement of sunshine-exposed branches, which implies that females could be also attracted by an increase in the release of some key stimulus from sunny leaves in oviposition [51,52]; further studies on this topic would be valuable. Through measuring and comparing tree heights, we found that the abundance of the available hostplants (i.e., AHPAb) has a positive interaction with butterfly occurrence, and it was also identified as the most critical driver (Table 4) of the HPL of *T. aureus*. On the contrary, a negative interaction was unexpectedly found between all hostplants’ abundance (HPAb) and butterfly occurrence, which may be due to the interference of unexposed trees; that is, the unavailable portion of hostplants does not match the butterfly’s early stages in distribution, perhaps due to a lack of stimulus for oviposition [51,52]. After selective felling, the abundance of tree seedlings in the secondary forests is even higher than that of the primeval forests [53] in the early recovery stage, and these seedlings are partially retained and subsequently take part in the process of restoration. However, most Magnoliaceae plants grow slowly [54]. Although the secondary forests have been in recovery for decades in the Jiulianshan Mountains [42], most of the hostplants still have their treetops unexposed and are not available as consumables. As such, the effect of HPL cannot be simply attributed to all hostplants (e.g., the HPAb), and the real contributor could be just the available portion (e.g., the HPAv).

*T. aureus* is a typical mountain forest species that often appears in the middle mountains above 1000 m at altitude. According to the Encounter-frequency hypothesis [55], this butterfly should be more inclined to select plants distributed in the middle mountains as host sources because of the higher encounter rate of these plants. Our results show that the hostplant resource in the middle-mountain areas is better in terms of diversity (e.g., MSD and HPSD) and abundance (e.g., MPAb and HPAb) than in the low mountain areas, and better availability (e.g., AHPSD and AHPAb) is assured in the middle mountains (Figure 4), which supports the above assumption. However, in addition to the effect of high elevation, another necessary condition is the habitat quality demanded by *T. aureus*.

In terms of habitat selection, the butterfly showed a “preference” for primeval forests, while showing “avoidance” of the low-quality habitats in secondary forests and coniferous forests in Bailey’s analysis (Table 2). Actually, only the primeval forests are able to simultaneously provide the highest species diversity or richness for the three known hostplants of the *T. aureus* canopy butterfly. The availability of hostplants (e.g., AHPAb) in the primeval forests was found to be significantly better than in secondary forests. However, it should be noted that when the availability was not considered, there was no significant difference between the primeval forest and secondary forest habitats in terms of the abundance of hostplants (e.g., HPAb and MPAb) (Figure 6). This means that the available portion truly represents the quality of hostplants (see the association between abundance and diversity in Figure 7) in a region, and the change in gradient reflected by this part was more than that of the habitat quality.

By comparing the occupied areas (OA) and non-occupied areas (NOA), we found some useful indicators reflecting the habitat quality demanded by *T. aureus*, including the tree age, height, and DBH, as well as the species diversity indices of the hostplant (e.g., MSD, HPSD, and AHPSD). However, the individual abundance of the hostplants (e.g., MPAb, HPAb, and AHPAb) could not be used as such indicators, because they did not differ between OA and NOA (Figure 5). A high hostplant abundance can only be contributed to a single hostplant species, which is not always consistent with a high-quality habitat. For instance, in our case study, we found that the *M. maudiae* hostplant was abundant in the OA and NOA, but the other two hostplant species, *M. foveolata* and *M. fordiana*, were rarely encountered in the NOA. Such asynchronism illustrates that the abundance index (e.g., HPAb, AHPAb) does not directly reflect the habitat quality, which means that the hostplant diversity is a better indicator of ‘resource-based habitat’ than the simple hostplant abundance [19]. Thus, it is necessary to simultaneously consider the habitat quality demanded by the butterflies when applying the concept of the “availability” of hostplants. Only in this way can we figure out the real contribution of the hostplant availability (e.g., AHPAb) to the HPL effect. We also suggest that it is important to pay more attention to the areas with high hostplant diversity in butterfly habitat conservation.

## 5. Conclusions

This case study in Jiulianshan clarifies that the forest canopy butterfly *T. aureus* prefers primeval forests, while avoiding other types of forest mainly due to two hierarchies of needs: (1) the basic hierarchy of needs, i.e., forest quality requirements, including elevation (e.g., middle-mountains), forest components (e.g., old-growth forests), forest structure (e.g., tree height and DBH of Magnoliaceae), and forest function (e.g., evergreen canopy), and (2) the advanced hierarchy of needs, especially the requirements for the key consumables of hostplants, including the hostplant species richness (e.g., MSD, HPSD, and AHPSD) and hostplant abundance (e.g., MPAb, HPAb, and AHPAb). Obviously, it is only after the basic needs are satisfied that the habitat can support the occurrence of this rare butterfly, and at this time, the HPL effect is then critically driven by the hostplant availability, rather than simply by the hostplant abundance alone. In a word, the severe requirements in terms of habitat and hostplant availability limit the occurrence of this canopy butterfly, which explains why, although the secondary forests in the area have been recovered for decades, the local population is still confined to a narrow, specific region, i.e., the last remaining primeval forests in Jiulianshan. Our results suggest that maintaining the quality of forest-based habitats with Magnoliaceae hostplants is critical for *T. aureus* conservation. This case study also has some useful implications for the conservation of other forest canopy butterflies.

## Figures and Tables

**Figure 1 insects-13-01082-f001:**
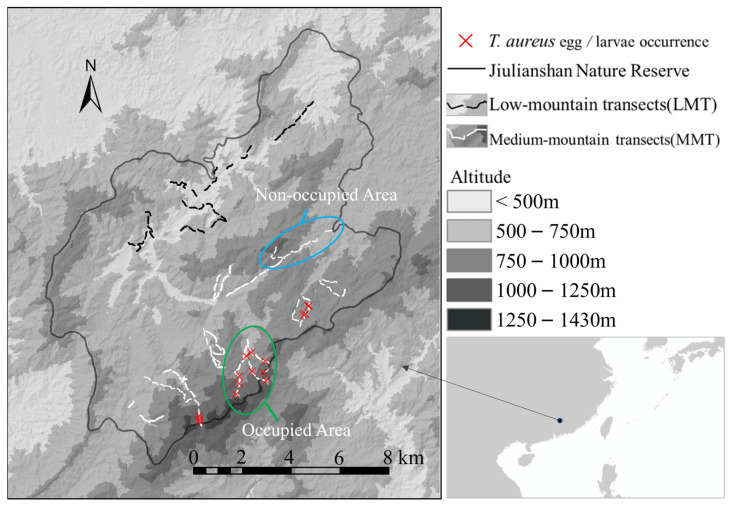
The geographical location of the study area, Jiulianshan National Nature Reserve (black line), and transect in low-mountain (black dotted line, with the top elevation < 1000 m, LMT) and middle-mountain (white dotted line, with the top elevation >1000 m, MMT) areas. The two ovals roughly indicate the butterfly occupied area (OA) and non-occupied area (NOA) for *T. aureus*.

**Figure 2 insects-13-01082-f002:**
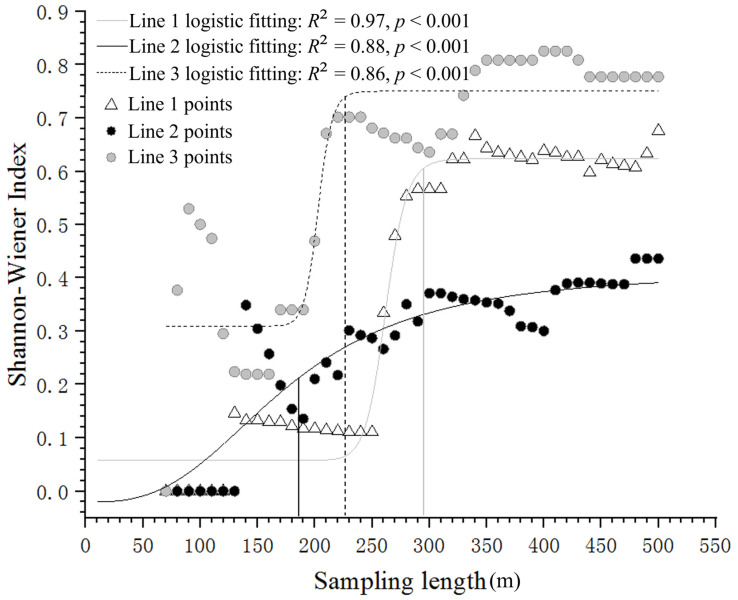
The logistic response curves fitted with the Shannon–Wiener index of Magnoliaceae plants against the transect length in the Jiulianshan Mountains, showing the points (e.g., 188 m, 226 m, and 295 m) of deceleration inflection in three transect lines extending from the mountain foot to the top.

**Figure 3 insects-13-01082-f003:**
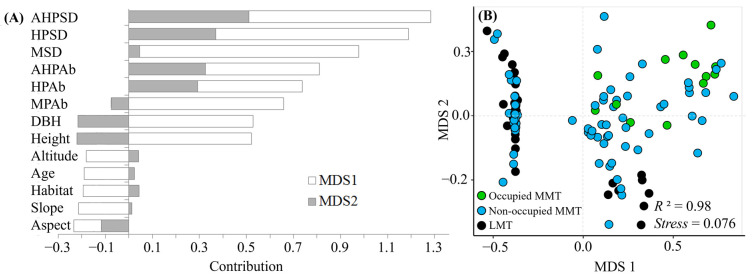
NMDS analysis of the transect sections, showing the contributions of environmental variables in Table 1 (**A**) and the distribution of the early butterfly stages of *T. aureus* occupied (Occupied MMT) or non-occupied sections of medium-mountain transects (MMT) as well as low-mountain transects (LMT) (**B**), see Table 1 for abbreviations.

**Figure 4 insects-13-01082-f004:**
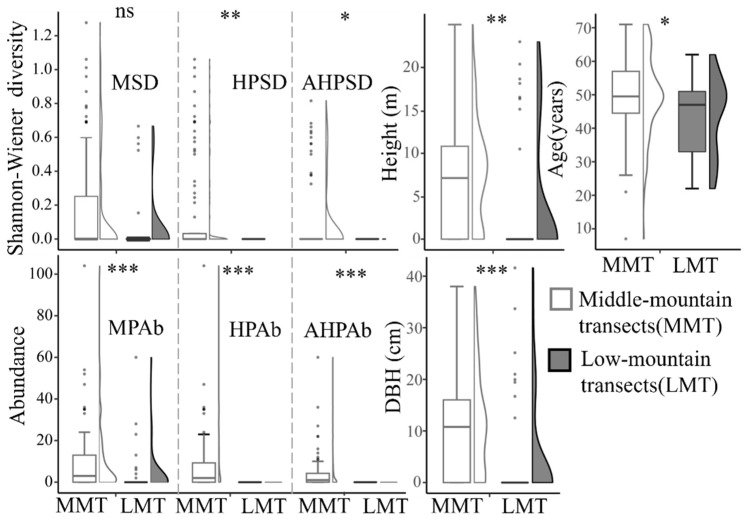
Comparison of DBH, tree height, tree age, plant abundance, and diversity in Magnoliaceae between low-mountain transects (LMT) and middle-mountain transects (MMT) in Jiulianshan. See Figure 1 for transects and Table 1 for abbreviations; ns for *p* > 0.05, * for *p* < 0.05, ** for *p* < 0.01, and *** for *p* < 0.001.

**Figure 5 insects-13-01082-f005:**
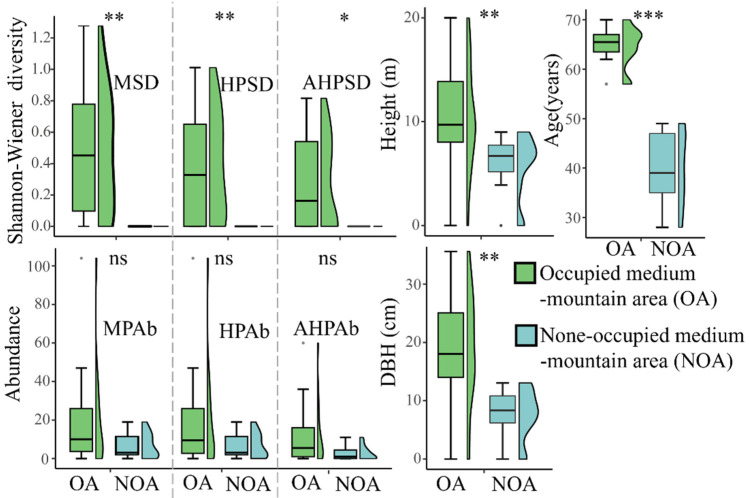
Comparison of DBH, tree height, tree age, plant abundance, and diversity in Magnoliaceae between the occupied and non-occupied areas of MMT in Jiulianshan. See Figure 1 for transects and Table 1 for abbreviations; ns for *p* > 0.05, * for *p* < 0.05, ** for *p* < 0.01, and *** for *p* < 0.001.

**Figure 6 insects-13-01082-f006:**
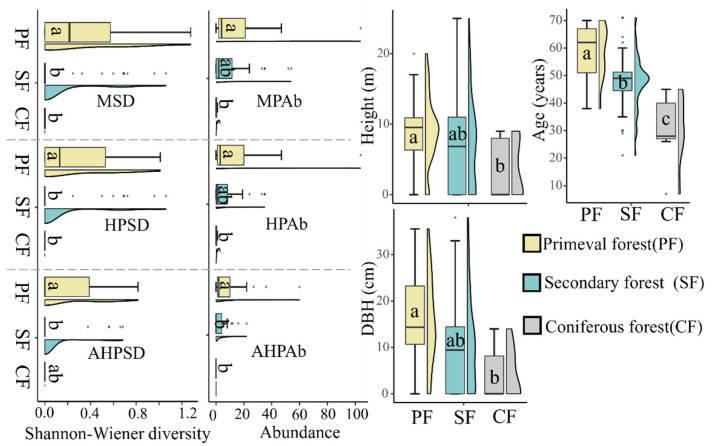
Comparison of the DBH, tree height, tree age, plant abundance, and diversity in Magnoliaceae among the primeval forest, secondary forest, and coniferous forest habitats in Jiulianshan. The different lowercase letters indicate significant differences between groups and vice versa. See Figure 1 for transects and Table 1 for abbreviations.

**Figure 7 insects-13-01082-f007:**
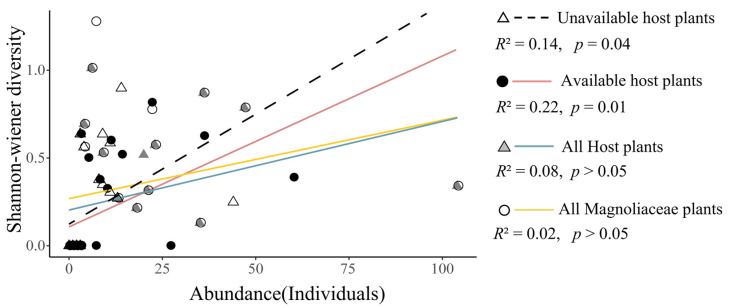
Abundance and diversity (Shannon–Wiener index) regression relationships for all Magnoliaceae plants, all hostplants, unavailable hostplants, as well as the available hostplants in primeval forests, showing the coefficients and *p* values. See Table 1 for abbreviations.

**Table 1 insects-13-01082-t001:** Environmental variables, abbreviations, and descriptions.

Environmental Variables	Abbreviations	Descriptions
Altitude	Altitude	the altitude of each transect section’s center point
Aspect	Aspect	the aspect of each transect section’s center point
Slope	Slope	the slope of each transect section’s center point
Average tree age	Age ^a^	the average tree age in each transect section
Diameter of breast height	DBH ^b^	the mean DBH of all Magnoliaceae trees in each transect section
Tree height	Height ^b^	the mean tree height of all Magnoliaceae trees in each transect section
All Magnoliaceae species diversity	MSD ^b^	the Shannon–Wiener index of all Magnoliaceae plants of each transect section
All Magnoliaceae plants abundance	MPAb ^b^	the individual number of all Magnoliaceae plants of each transect section
All hostplants species diversity	HPSD ^b^	the Shannon–Wiener index of all hostplants of each transect section
All hostplants abundance	HPAb ^b^	the individual number of all hostplants of each transect section
Available hostplants species diversity	AHPSD ^b^	the Shannon–Wiener index of the available hostplants with the height above the average tree height of each transect section
Available hostplants abundance	AHPAb ^b^	the individual number of the available hostplants with the height above the average tree height of each transect section
Habitat type	Habitat	the habitat of each transect section, including primeval, secondary, or coniferous forest

^a^ Sourced from the local database of Forest Resource Inventory (FRI), the average forest tree age is determined by the dominant tree species; ^b^ Without considering the seedling individuals with DBH < 3 cm.

**Table 2 insects-13-01082-t002:** Habitat selection of *T. aureus* by the method of Bailey’s.

Habitat Types	Actual Proportion Utilized (P_i_)	Expected Proportion Utilized (P_io_)	Bailey’s 95% Confidence Interval for P_i_	Selection
Primeval forests	0.8462	0.2101	0.4372 ≤ P_i_ ≤ 0.9847	Preference
Secondary forests	0.1538	0.6807	0.0021 ≤ P_i_ ≤ 0.4954	Avoidance
Coniferous forests	0	0.1092	0.0000 ≤ P_i_ ≤ 0.0000	Avoidance

**Table 3 insects-13-01082-t003:** Comparisons of environmental variables between occupied and non-occupied transect sections in *T. aureus*.

Environmental Variables	Non-Occupied (*n* = 106)	Occupied (*n* = 13)	*p* Value
Altitude	662.68 ± 168.1 ^b^	983.00 ± 154.64 ^a^	<0.001 ***
Aspect	0.52 ± 0.31 ^a^	0.34 ± 0.25 ^a^	>0.05 ns
Slope	16.83 ± 8.78 ^b^	17.54 ± 6.09 ^a^	>0.05 ns
Age	46.46 ± 11.77 ^b^	60.15 ± 10.54 ^a^	<0.001 ***
DBH	8.43 ± 10.50 ^a^	17.51 ± 7.72 ^a^	<0.01 **
Height	5.81 ± 6.91 ^b^	9.80 ± 3.11 ^a^	<0.01 **
MSD	0.11 ± 0.27 ^b^	0.41 ± 0.37 ^a^	<0.001 ***
MPAb	6.08 ± 11.70 ^b^	24.23 ± 27.94 ^a^	<0.001 ***
HPSD	0.07 ± 0.22 ^b^	0.36 ± 0.32 ^a^	<0.001 ***
HPAb	3.35 ± 7.33 ^b^	23.85 ± 28.13 ^a^	<0.001 ***
AHPSD	0.04 ± 0.15 ^b^	0.29 ± 0.30 ^a^	<0.001 ***
AHPAb	1.46 ± 3.48 ^b^	15.15 ± 17.33 ^a^	<0.001 ***

The different lowercase letters indicated significant differences between occupied and non-occupied sections and vice versa. ns for *p* > 0.05, ** for *p* < 0.01 and *** for *p* < 0.001. See Table 1 for abbreviations.

**Table 4 insects-13-01082-t004:** Discriminant coefficient and Wilk’s λ results for the four selected environmental variables included in the stepwise discriminant analysis.

Environmental Variables	Standardized Discriminant Coefficient	Wilk’s λ	*p*
AHPAb	1.464	0.613	<0.001
Altitude	0.460	0.592	<0.001
Habitat	0.334	0.572	<0.001
HPAb	−0.924	0.571	<0.001

See Table 1 for abbreviations.

## Data Availability

Data can be provided on request from the corresponding author.

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
