# Peer review of "Forest Quality and Available Hostplant Abundance Limit the Canopy Butterfly of Teinopalpus aureus"

_insects, 2022, doi:10.3390/insects13121082_

Round 1
Reviewer 1 Report
A generally high quality paper on an important conservation ecological issue. Minor revision is suggested based on the "bubbles" attached to the text.
The main suggestions are as folllows:
Row 45 - An open question of wording: herbivorous vs phytophagous. This occasion I would like to prefer "Phytophagous". At first: Magnolia spp. are trees or scrubs, but not herbs. At second: the term phytophagous forms a "family" of terms together with oligophagous, polyphagous, monophagous, etc.
Rows 56-58 - I questioned the logical structure of the sentence. If an oligophagous insect is limited by its host plant, how to separate the abiotic limitatition of the butterfly from the abiotic limitation of its hostplant? Theoretically OK, but in field survey???
Row 78 - The References 27-28 are for me superfluous! These European butterflies are not well comparable with the large Papilionid butterflies neither in their behaviour nor in habitat limitations (these are even rather different also in the cases of Ch. briseis vs. Haemaris lucina).
Rows 90ff and132ff - In the general characterisation of the target species T. aureus (see: rows 90ff) or at least in the last paragraph of the Introduction (132ff) it should be mentioned that this species has two generations pro year!
Row 215 - A question of wording: "behavior" is the form of the American English? What do you prefer? Oxford vs American or vice versa.
Rows 332ff - Some additionarry suggestions to the Discussion.
(i) Since the Magnoliaceae are one of the phylogenetically most ancient Angiospermae, and they are also rich in different volatiles (many references!), thus this co-evolved food-plant specialisation of the phylogenetic relict butterflies Taeniopalpus spp. to Magnoliaceae must be a phylogenetically ancient connection, as well. Therefore the conservation priority of this species must be exceptionally high also in evolutionary biological respect!
(ii) Of course, it would be important to know whether the volatile content of the "sunny" leaves could influence the oviposition of the butterfly as a key stimulus?!
Row 353 - Question: perhaps the volatile content of sunny leaves is functioning as a key stimulus? Observation vs neglection!
Row 360 - According to a possible preliminary working hypothesis (see above) it may be connected with some chemical ecological factors.
Rows 368ff - There are much more references on grassland butterflies which show that there are groups mostly limited by food plant specialisation (e.g. Lycaenidae developing on poisonous Fabaceae) vs. not food-plant specialist Satyrinae limited by the grassland structure. In such cases the key signals of oviposition may also be different.
Row 403 - T. aureus has two generations in a vegetation period. Thus the asynchronism may be different according to saisons.
If the Authors disagree with some of my sentences, I am usually ready for open discussions.

Author Response
Response to Reviewer 1 Comments
Dear reviewing expert,
Many thanks for the comments on our manuscript “Forest quality and available hostplant abundance limits the canopy butterfly, Teinopalpus aureus, from a case of Jiulianshan, South China”. We appreciate and accept the modification suggestions and have revised the manuscript accordingly. The revised parts are marked up using the “Track Changes” function in the revision version. The detailed responses to the reviewer’s comments are presented as follows:
Point 1: Row 45 - An open question of wording: herbivorous vs phytophagous. This occasion I would like to prefer "Phytophagous". At first: Magnolia spp. are trees or scrubs, but not herbs. At second: the term phytophagous forms a "family" of terms together with oligophagous, polyphagous, monophagous, etc.
Response 1: Thanks to the suggestion, we follow this suggestion and use “phytophagous” in this revision version.
Point 2: Rows 56-58 - I questioned the logical structure of the sentence. If an oligophagous insect is limited by its host plant, how to separate the abiotic limitatition of the butterfly from the abiotic limitation of its hostplant? Theoretically OK, but in field survey???
Response 2: Thank you reminding us, this sentence is modified as "...they are more likely to be limited together by abiotic (climate, topography, etc.) and biotic factors (e.g. the bottom-up control by hostplants).".
Point 3: Row 78 - The References 27-28 are for me superfluous! These European butterflies are not well comparable with the large Papilionid butterflies neither in their behaviour nor in habitat limitations (these are even rather different also in the cases of Ch. briseis vs. Haemaris lucina).
Response 3: Yes, we agree that, and now we use another two forest butterflies, Byasa impediens [25] and Bhutanitis thaidina [26], as examples to emphasis the importance of favor microhabitat in occurrence of forest butterflies.
[25] Li, X.; Zhang, Y.-L.; Settele, J.; Franzén, M.; Schweiger, O. Long-Distance Dispersal and Habitat Use of the Butterfly Byasa Impediens in a Fragmented Subtropical Forest and 4 The Station of Forests Pests and Diseases Control and Quarantine. Insect Conservation and Diversity 2012, 6, doi:10.1111/j.1752-4598.2012.00199.x.
[26] Gao, K.; Li, X.; Guo, Z.; Zhang, Y. The Bionomics, Habitat Requirements and Population Threats of the Butterfly Bhutanitis Thaidina in Taibai Mountain. J Insect Conserv 2014, 18, 29–38, doi:10.1007/s10841-014-9612-1.
Point 4: Rows 90ff and132ff - In the general characterisation of the target species T. aureus (see: rows 90ff) or at least in the last paragraph of the Introduction (132ff) it should be mentioned that this species has two generations pro year!
Response 4: Thanks for reminding, we supplement "... two generations pro year..." in the last paragraph of the introduction, please check the revision version.
Point 5: Row 215 - A question of wording: "behavior" is the form of the American English? What do you prefer? Oxford vs American or vice versa.
Response 5: Thanks for reminding, we use “behaviour” instead.
Point 6: Rows 332ff - Since the Magnoliaceae are one of the phylogenetically most ancient Angiospermae, and they are also rich in different volatiles (many references!), thus this co-evolved food-plant specialisation of the phylogenetic relict butterflies Teinopalpus spp. to Magnoliaceae must be a phylogenetically ancient connection, as well. Therefore the conservation priority of this species must be exceptionally high also in evolutionary biological respect!
Of course, it would be important to know whether the volatile content of the "sunny" leaves could influence the oviposition of the butterfly as a key stimulus?!
Response 6: Great, we also have the same idea that there could be a phylogenetical ancient connection between this oligophagous butterfly and Magnoliaceae, one of the most ancient Angiospermae on the earth. And obviously, some further research works are necessary and significant, especially those focusing on the volatile content influencing the oviposition of Teinopalpus spp., T. aureus and T. imperialis.
Point 7: Row 353 - Question: perhaps the volatile content of sunny leaves is functioning as a key stimulus? Observation vs neglection!
Response 7: Yes, we agree with that, and further studies will be valuable, such as to investigate the possible stimulus of volatile in butterfly oviposition.
Point 8: Row 360 - According to a possible preliminary working hypothesis (see above) it may be connected with some chemical ecological factors.
Response 8: Yes, we accept the hypothesis gladly, and we will devote to provide chemical ecological evidences in future studies.
Point 9: Rows 368ff - There are much more references on grassland butterflies which show that there are groups mostly limited by food plant specialisation (e.g. Lycaenidae developing on poisonous Fabaceae) vs. not food-plant specialist Satyrinae limited by the grassland structure. In such cases the key signals of oviposition may also be different.
Response 9: Yes, we did consult many references on grassland butterfly's oviposition, habitat selection, etc.. However, as you pointed out, the situation could be not the same between grassland and forest butterflies, since the integrity of forest structure is usually vital to many species, especially the large swallowtail butterflies. Our case of T. aureus showed that the habitat selection of this canopy butterfly was totally driven by the high hostplant diversity and availability. And such conditions can only be satisfied by the high-quality primeval forests in Jiulianshan. Therefore, we deleted several literatures (on grassland butterfly) with low correlation with our case, and also supplemented two literatures related (on forest butterfly), please check "Reference".
Point 10: T. aureus has two generations in a vegetation period. Thus the asynchronism may be different according to seasons.
Response 10: Great, there may be asynchrony in hostplant or habitat selection between the two generations. This assumption is also valuable to prove in future studies. Here, the asynchrony proposed purely referred to the spatial mismatch between hostplant diversity and hostplant abundance (individual). For example, the abundance in secondary forests seemed as high as that in primeval forests, but actually there are only one single hostplant species occurred in most secondary forests, which is totally different from the situation in primeval forests.
Sincerely,
All authors
Reviewer 2 Report
Comments on Forest quality and available hostplant abundance limits the canopy butterfly, Teinopalpus aureus, from a case of Jiulianshan, South China
General comments
Overall, this is an interesting article. While the authors have conducted an excellent job in conducting this research, it’s a bit difficult to follow due to the sentence/paragraph structure. I think this study will attract readers of insects journal.
Specific comments
From my perspective, the title is a bit long. Maybe shorten it? You can remove everything after ‘canopy butterfly’.
Simple Summary
Good summary
Abstract
Good summary
Please try not to use any word in the keyword section that’s already in the title.
The introduction section should look like a coherent story; otherwise, it’s really difficult to follow. For example, some sentences are quite big and not connected to each other. Could you please make it easier for the readers? Beside, given the field study was conducted in a nature reserve, it would be great if the authors could write a few sentences on protected areas and insect conservation. Here, this recent review on insect conservation would be of help: https://doi.org/10.1016/j.tree.2022.09.004.
Chowdhury, S., Jennions, M. D., Zalucki, M. P., Maron, M., Watson, J. E., & Fuller, R. A. (2022). Protected areas and the future of insect conservation. Trends in Ecology & Evolution.
Line 91: ‘one of the largest’
Method section looks okay and easy to follow!
Figure 3A. Please review the figure by moving the levels further left, so that the patterns become easier to understand.
Figure 6/7. Please change the colour ‘green’ to other.
Author Response
Response to Reviewer 2 Comments
Dear reviewing expert,
Many thanks for the comments on our manuscript “Forest quality and available hostplant abundance limits the canopy butterfly, Teinopalpus aureus, from a case of Jiulianshan, South China”. We appreciate and accept the modification suggestions and have revised the manuscript accordingly. The revised parts are marked up using the “Track Changes” function in the revision version. The detailed responses to the reviewer’s comments are presented as follows:
Point 1: From my perspective, the title is a bit long. Maybe shorten it? You can remove everything after ‘canopy butterfly’.
Response 1: Thanks to the suggestion, we partly follow this suggestion and change the title as “Forest quality and available hostplant abundance limit the canopy butterfly of Teinopalpus aureus”. Of course, any further suggestions will be welcomed.
Point 2: Please try not to use any word in the keyword section that’s already in the title.
Response 2: Thanks to the suggestion, we delete the keyword of Teinopalpus aureus which is totally repeated the title.
Point 3: The introduction section should look like a coherent story; otherwise, it’s really difficult to follow. For example, some sentences are quite big and not connected to each other. Could you please make it easier for the readers?
Response 3: Many thanks. Yes, it is our responsibility to make it easier for the readers. This time, we firstly checked the English in introduction, where several big sentences were re-written accordingly. As for other parts, we also made some modifications, including sentence, grammar mistakes, capitalization matters etc.. Secondly, we asked two colleagues for English revision, after sending them the manuscript in email. Therefore, there are many changes in this revised version, and the English is improved greatly. Obviously, the English can be further improved if spending more time. But we have to upload the revised file together with cover letter by 7 November demanded in the system, so further modifications of English will be done next time, if needed, such as asking the professional organization for polishing. Thanks again!
Point 4: Beside, given the field study was conducted in a nature reserve, it would be great if the authors could write a few sentences on protected areas and insect conservation. Here, this recent review on insect conservation would be of help: https://doi.org/10.1016/j.tree.2022.09.004. Chowdhury, S., Jennions, M. D., Zalucki, M. P., Maron, M., Watson, J. E., & Fuller, R. A. (2022). Protected areas and the future of insect conservation. Trends in Ecology & Evolution.
Response 4: Thanks, we wrote a few sentences on protected areas and butterfly conservation, and added them in the part of introduction, discussion and conclusion respectively. These sentences are as follows:
“As such, strategically establishing their key habitat as a protection area is a basic element to conserve the populations [18].” (in introduction)
“Also, it suggests that it is valuable to pay more attentions on the areas with high diversity of hostplant in butterfly habitat conservation.” (in discussion)
“Our results suggest that maintaining the quality of forest-based habitat with Magnoliaceae hostplants was critical in T. aureus conservation.” (in conclusion)
[18] Chowdhury, S.; Jennions, M.D.; Zalucki, M.P.; Maron, M.; Watson, J.E.M.; Fuller, R.A. Protected Areas and the Future of Insect Conservation. Trends in Ecology & Evolution 2022, 0, doi:10.1016/j.tree.2022.09.004.
Point 5: Figure 3A. Please review the figure by moving the levels further left, so that the patterns become easier to understand.
Response 5: Ok, the figure position has been moved accordingly.
Point 6: Figure 6/7. Please change the colour ‘green’ to other.
Response 6: Ok, we redrew the two figures, and modified the colour accordingly.
Sincerely,
All authors